# Artificial Neural Networks for Predicting Mechanical Properties of Crystalline Polyamide12 via Molecular Dynamics Simulations

**DOI:** 10.3390/polym15214254

**Published:** 2023-10-29

**Authors:** Caglar Tamur, Shaofan Li, Danielle Zeng

**Affiliations:** 1Department of Civil and Environmental Engineering, University of California, Berkeley, CA 92740, USA; caglar.tamur@berkeley.edu; 2Ford Motor Company, Dearborn, MI 48126, USA; dzeng@ford.com

**Keywords:** molecular dynamics, crystalline polymers, ReaxFF, polyamide, machine learning, neural network

## Abstract

Predicting material properties of 3D printed polymer products is a challenge in additive manufacturing due to the highly localized and complex manufacturing process. The microstructure of such products is fundamentally different from the ones obtained by using conventional manufacturing methods, which makes the task even more difficult. As the first step of a systematic multiscale approach, in this work, we have developed an artificial neural network (ANN) to predict the mechanical properties of the crystalline form of Polyamide12 (PA12) based on data collected from molecular dynamics (MD) simulations. Using the machine learning approach, we are able to predict the stress–strain relations of PA12 once the macroscale deformation gradient is provided as an input to the ANN. We have shown that this is an efficient and accurate approach, which can provide a three-dimensional molecular-level anisotropic stress–strain relation of PA12 for any macroscale mechanics model, such as finite element modeling at arbitrary quadrature points. This work lays the foundation for a multiscale finite element method for simulating semicrystalline polymers, which will be published as a separate study.

## 1. Introduction

Material modeling has been a great challenge for additive manufacturing and other advanced manufacturing technologies. In additive manufacturing, the product is built layer by layer; thus, the resultant properties are highly dependent on the printing process both spatially and temporally at the localized regions. A microstructurally informed and robust mechanical model is highly anticipated, as it would help to overcome the challenges with manufacturing process, product design, and material selection.

One promising approach is multiscale modeling of 3D printed materials [1,2,3], which has the ability to extract bottom-up material information to assess the macroscale properties of the product. However, the cost of multiscale modeling may vary, especially when it is involved with molecular-scale simulations. Traditionally, in multiscale modeling of crystalline materials, this is overcome by using a technique called the Cauchy–Born rule [4,5], which is an approximation of molecular dynamics by using simplified molecular statics or by using the phase field crystal method for polycrystalline metals [6]. However, for semicrystalline or amorphous materials, due to the lack of a definite crystal structure, full-scale molecular dynamics must be utilized, which will significantly increase the cost of the multiscale simulation. This is because the macroscale finite element method would require a molecular-level stress–strain relation at each quadrature point for an arbitrarily given strain and it is almost impossible to conduct concurrent molecular dynamics simulations unless one has abundant computational resources.

The task at hand is to simulate the molecular-level stress–strain relation under arbitrary strain and temperature with good accuracy, efficiency, and low cost. To solve this problem, in the present work, we adopted a machine learning approach by developing an artificial neural network (ANN) that is trained on a molecular dynamics (MD) data set, which can predict the molecular-level mechanical material properties of polymeric materials.

Current efforts in machine learning-driven material property predictions involve a variety of approaches. One common approach is to use features based on the material composition to obtain structure and electronic properties, such as predicting the band gap of semiconductor materials using convolutional neural networks [7] and predicting adsorption energies of high-entropy alloys through deep neural networks [8]. In a recent study, standard feedforward neural networks are used with EAM-based MD simulations to model crystal plasticity in a multiscale framework [5].

Additive manufacturing can use a wide range of materials, such as synthetic polymers, metal alloys [9], and biomaterials. Industrial applications range from the automotive industry to tissue engineering [10], and understanding the mechanical properties of such materials is still an open topic. Our focus in this work is on polymers, for which new applications emerge everyday, such as polyester membranes for biomedical applications [11] and geopolymers used in the construction industry [12].

Polymeric materials have three different microstructural forms: crystal, amorphous, and semicrystalline. As a starting point, we first develop an ANN to predict the mechanical response of the anisotropic crystalline forms of polymers. The approach we presented can be extended to include amorphous phases and will eventually be incorporated into the constitutive relations of a multiscale finite element method for semicrystalline polymers. One of the polymeric materials widely used in additive manufacturing is Polyamide12 (PA12), also known as Nylon12, which is a synthetic polymer that is used in many industrial applications, such as automotive parts, aerospace applications, and medical components. An automotive part manufactured using the multi-jet fusion process is shown in Figure 1.

PA12 has the chemical formula [-(CH_2_)_11_C(O)NH-]_*n*_ as visualized in Figure 2. In recent years, many 3D printed PA12 products have been fabricated and they have shown some outstanding material properties. Additively manufactured PA12 is a semicrystalline material, in which the microstructure has a sandwich-like structure of alternating regions of crystal and amorphous zones.

Its structural characterization has been extensively studied since the 1970s with X-ray diffraction (XRD) experiments [13,14] and with nuclear magnetic resonance (NMR) spectroscopy [15], which revealed that PA12 displays polymorphism and can have crystal phases $\alpha ,\alpha^{\prime },\gamma$, and $\gamma^{\prime }$. The most abundant phase is the γ form, which results from slow cooling at atmospheric pressure, whereas the other phases require specific conditions such as rapid quenching and/or high-pressure treatment. For the purposes of our study, we focus on the γ crystal form.

## 2. Molecular Dynamics Simulations

In this section, we describe the details of molecular dynamics simulations, such as the preparation of initial configurations, force field selection, and simulation procedures, and we conduct a preliminary study.

### 2.1. System Setup

We start by constructing the unit cell of the γ form PA12 crystal. The γ phase has a pseudo-hexagonal monoclinic structure with the lattice parameters summarized in Table 1 [13,14]. Using the atomistic coordinates and the lattice parameters from the experimental literature [14], the unit cell of the γ phase is constructed. This monoclinic structure contains four PA12 chains and is visualized in Figure 3. Note that the dashed lines indicate hydrogen bonds and that the unit cell is periodic in all directions.

Polymer chains are aligned with the y-axis, and the unit cell represents a perfect crystalline system with infinite chain length and no defects. MD simulations of triclinic (non-orthogonal) boxes are computationally expensive, due to the irregular partitioning of the processor subdomain. To improve the efficiency, we transform the simulation box into an orthogonal cell with chains aligned with the z-axis, which is depicted in Figure 3. The unit cell is then duplicated by (8×4×2) in the (x, y, z) directions, respectively, to create a supercell for MD simulations. The resulting model, which consists of around 9500 atoms and has dimensions [38.3×33.2×63.8], is visualized in Figure 4.

### 2.2. Force Field Selection

Force fields represent the interactions between atoms and molecules using a set of equations. In classical forms, force fields consist of empirical potentials that describe bonded interactions (primary bonds, bond angles, and dihedrals) and non-bonded interactions (van der Waals and electrostatic). Choosing a suitable force field for the system under consideration is a crucial part of molecular dynamics simulations.

Several empirical force fields have been used in the simulation of polyamides, some popular examples being OPLS, CVFF, COMPASS, and DREIDING [16]. However, these potentials are typically unable to model chemical reactions and cannot account for bond dissociation. Hence, they are not the ideal choice for investigating the mechanical response of the system in the context of fracture mechanics and rupture of polymer crystals, which may involve the chain scission process [17,18].

Reactive force fields, on the other hand, have been developed to simulate complex chemical reactions and have been used successfully in deformation simulations that involve bond cleavage. Once such a force field is ReaxFF, originally developed by van Duin [19], which utilizes the so-called bond order formalism to describe chemical bonding. ReaxFF requires no topology information and it is easy to construct initial MD configurations since only the atomistic coordinates are needed. ReaxFF specially treats weak hydrogen bonds with an explicit energy term in the functional, which plays an important role in polymer systems. The current implementation in LAMMPS utilizes the functional described in [20].

ReaxFF is parameterized for a wide range of materials through quantum mechanical calculations [21] and has also been used in the simulation of polyamide crystals before [22]. For our polymer system, we tested three such parameterizations [23,24,25]. The test process involved relaxing the initial structure in the NPT ensemble and comparing the resultant lattice parameters with the experimental values. Among these, the ReaxFF parameterization by Mattson [24] yielded the best result; therefore, it was chosen as the force field for this study.

### 2.3. Simulation Details and Preliminary Analysis

Molecular dynamics simulations are performed using the Large-Scale Atomic/Molecular Massively Parallel Simulator (LAMMPS) [26]. The simulations are run on Berkeley’s high-performance computing cluster Savio using a [4×2×4] CPU grid. MD time steps are chosen as 0.5 fs. Nosé–Hoover thermostat and barostat [27] are used to control temperature and pressure, respectively, with damping parameters chosen as 50 fs and 500 fs. Damping parameters were chosen according to the LAMMPS documentation, which recommends temperature and pressure damping to be 100 and 1000 times the time step, respectively.

The initial structure is relaxed in the constant pressure and temperature (NPT) ensemble at 300 K and 1 atm, where the pressure is controlled independently in all directions. We observed that an NPT simulation for 50 ps was sufficient to reach equilibrium, as the lattice parameters stabilized adequately at this point.

As a preliminary study and to investigate in detail the anisotropic mechanical response of PA12 crystals, we constructed a larger MD cell with 38,000 atoms. After relaxation under ambient conditions, we deform the system in different directions under uniaxial tension. Uniaxial deformation is imposed by stretching the unit cell in one direction at a constant strain rate and relaxing the other two dimensions with the NPT ensemble at 1 atm. For this part of the study, we used slower strain rates and deformed the cell quasi-statically in a step-by-step fashion as described in [28]. In this process, the MD cell is deformed to 1% of its final stretch and relaxed in the NVT ensemble, and the stress is sampled and averaged over 5 ps intervals. These steps are repeated until rupture of the polymer chains is observed and the resulting stress–strain behaviors are shown in Figure 5 for each direction.

In Figure 5, we observe that the system is highly anisotropic; crystal is significantly more stiff in the z-direction, which is the direction of polymer chain alignment, and it is more ductile in the other two directions. Interestingly, for the x-direction, Figure 5a, there was a remarkable increase in the elastic modulus once the strain reached around 12%. We explored this phenomenon to comprehend whether it involves a mechanically induced phase transformation, by deforming the original monoclinic crystal supercell in the x-direction. The resulting atomistic configurations before and after the change in elastic modulus are visualized in Figure 6, in which we clearly observe a change in the microstructural configurations.

Upon additional analysis, we understood that the process is completely reversible and the effect disappears once the MD cell is unloaded to its original state. Further investigation may be required to draw a meaningful conclusion about the phenomenon.

## 3. Artificial Neural Network for Constitutive Law

In this section, we present a data-driven approach to model the hyperelastic constitutive law of crystalline PA12, to be used in multiscale mechanics simulations. Our task is to find the mapping between the deformation state and the resulting material response, using the data obtained from the molecular dynamics simulations on an artificial neural network (ANN). We discuss data collection, model selection, training phase, and predictions in detail.

### 3.1. Data Collection and Processing

In order to generate the data set to train the learning model, we performed MD simulations by deforming the PA12 supercell in different directions with varying strain rates. For the purposes of this study, we limit our attention to uniaxial and biaxial tensile deformations, although the presented methodology should be applicable to any deformation mode. Using the findings of the preliminary study, Figure 5, we determined the final stretch limits of the simulation box necessary in each direction and corresponding strain rate ranges, as summarized in Table 2. In all simulations, the supercell is deformed for 10,000 steps with a rate chosen from the prescribed ranges depending on the loading direction. Specifically, we chose 15 evenly spaced values from the ranges in Table 2, and took their combinations to obtain uniaxial and biaxial loading cases, resulting in 720 different MD simulations, each having different final stretches and strain rates. Note that the ultimate strains would only be reached at simulations which utilize the maximum strain rates.

Each deformation simulation is run for 5 ps, and the virial stress and the box dimensions are sampled at 50 fs intervals. Consequently, after running all the 720 simulations, we collected a data set consisting of 72,720 data points. Each data point is 12 dimensional, representing the deformation (six components) and stress (six components) states of the PA12 crystal. Note that the process described here involves remarkably high strain rates and short simulation times, since it was not computationally feasible to conduct slower simulations and collect a large data set at the same time.

The original data from MD simulations consist of box dimensions $l=\{l_{x},l_{y},l_{z}\}$ which can be used to construct the deformation gradient F, and the stress measure we get is the pressure tensor, known as the virial stress. Virial stress has been shown to be equivalent to continuum Cauchy stress σ [29]. In order to adopt the model into the continuum mechanics framework, we need to transform the data set into the tensor quantities utilized in finite deformation. To preserve material objectivity [5,30], we chose the energetic conjugates right Cauchy–Green tensor C and the second Piola–Kirchhoff (PK2) stress tensor S to represent the deformation state and material response, respectively. Each of these second-order tensors have nine components, but making use of the symmetry and Voigt notation we represent them in the vector form as below.
(1)$$\begin{array}{cc} C & =\left[C_{11},\;C_{22},\;C_{33},\;C_{12},\;C_{13},\;C_{23}\right] \end{array}$$
(2)$$\begin{array}{cc} S & =\left[S_{11},\;S_{22},\;S_{33},\;S_{12},\;S_{13},\;S_{23}\right] \end{array}$$

Using the elementary relations of continuum mechanics, we can obtain the C and S tensors as follows.
(3)$$\begin{array}{cc} J & =\det F \end{array}$$
(4)$$\begin{array}{cc} C & =F^{T}F \end{array}$$
(5)$$\begin{array}{cc} S & =JF^{-1}\sigma F^{-T} \end{array}$$

Now, we can state our goal as finding the map,
(6)Ψ:C→S
where Ψ encapsulates the constitutive model that we are going to approximate through supervised learning.

Finally, we normalize our input data set to have a mean of zero and a standard deviation of one, a process known as *standardization*. It is common practice in gradient-based learning methods to standardize the data, which improves the performance of ANNs by helping to solve the convergence issues of the backpropagation algorithm [31,32]. Standardization can be described as follows.
(7)$$C_{ij}^{std}=\frac{C_{ij}-{\hat{\mu }}_{ij}}{{\hat{\sigma }}_{ij}}\;\;\;\mathrm{for}\;\;\;i,j=1,2,3.$$
where $C_{ij}^{std}$ is the normalized component of the right Cauchy–Green tensor, ${\hat{\mu }}_{ij}$ is the sample mean, and ${\hat{\sigma }}_{ij}$ is the sample standard deviation of the respective component.

### 3.2. Model Selection and Results

We adopt a fully connected feedforward network architecture to construct our regression ANN, which may be called a multilayer perceptron (MLP). As shown by the universal approximation theorem, feedforward neural networks can approximate any continuous function, provided that they have at least one hidden layer, have enough neurons, and the activation functions satisfy certain properties [33,34]. Thus, we believe that an MLP is a suitable choice to approximate the hyperelastic constitutive law. The Keras [35] Python interface of the TensorFlow [36] machine learning package is used to select and train our ANN model.

Our task is to find an approximation to the constitutive model defined in Equation (Equation 6). Formally, we can express the learning problem as follows.

Find
(8)$$N\left(C,w\right)=\hat{S}$$
such that
(9)$$w=\underset{w}{\mathrm{argmin}}L\left(S,\hat{S}\right)$$
where N encodes the ANN, w is known as the *weight* of each neuron, and L is a *loss function*. The task in supervised learning is to find optimal weights w such that the metric defined in the loss function is minimized.

To select the ANN model parameters known as *hyperparameters*, such as the number of hidden layers, the number of neurons in each layer, the type of activation functions, and the learning rate of the gradient descent, we perform hyperparameter optimization. We chose two candidate activation functions that are commonly used in regression, the *ReLu* function and its smooth approximate version known as the *Softplus* function.
(10)$$\mathrm{ReLu}\;:\;\sigma \left(x\right)=\max\left(x\right)\;\;\;\;\;\;\;\;\;\mathrm{Softplus}\;:\;\sigma \left(x\right)=\log\left(1+e^{x}\right)$$

For the loss function, we choose the mean squared error (MSE) of the PK2 stress, which is defined as
(11)$$L_{\mathrm{MSE}}\left(S,\hat{S}\right)=\frac{1}{n}\sum_{i=1}^{n}\left|\right|S_{i}-{\hat{S}}_{i}{\left|\right|}_{2}^{2}\;,$$
where S is the PK2 stress obtained from MD simulations, $\hat{S}$ is the predicted stress from ANN, and *n* is the number of data points.

We tried two methodologies to select the best set of hyperparameters, the HyperBand algorithm [37] and Bayesian Optimization with Gaussian Process [38]. Both algorithms would pick the best parameters from a predefined set, which would give the lowest validation loss. The set of parameters that we search for is determined by preliminary analysis, where the optimizer of the ANN is chosen as the Adam algorithm [39], which implements a version of the stochastic gradient descent (SGD) method. Accordingly, we chose the range of parameters to perform hyperparameter tuning, as shown in Table 3.

We start by partitioning the data into a random 80–20% train–test split. Then, we run hyperparameter optimization using Hyperband and Bayesian Optimization methods in the domain described in Table 3, leaving aside 20% of the training data to be used to compute validation loss. For all models, the learning rate $\gamma =10^{-2}$ and the number of hidden layers of four or five gave us the best results. For the rest of the hyperparameters, we investigated the top ten models and chose the ones with the lowest complexity to reduce overfitting. Resulting candidate models are summarized in Table 4.

The models summarized above are trained on the train data set for 1000 epochs and predictions are made on the test data to assess our final performance. The resulting train-test history curves, in terms of MSE in PK2 stress (MPa), are presented in Figure 7. We conclude that the ANNs are stable, there is no significant overfitting, and they perform well against the test data set, as seen from the test loss. The best model appears to be ANN-4, the rightmost column in Table 4, and its architecture is schematically visualized in Figure 8.

Finally, we tested ANN-4 against a new set of uniaxial tension MD simulations to observe the performance of our models against the variance in the strain rate. Two of the simulations utilized strain rates ${\dot{\epsilon }}_{1}=5\times 10^{-6}/fs$ and ${\dot{\epsilon }}_{2}=10\times 10^{-6}/fs$, which are within the training interval defined in Table 2. Predictions from ANN-4 are compared with the MD results, and the resultant $C_{33}$ versus $S_{33}$ (MPa) relations are shown in Figure 9. The predictions in the elastic range and the ultimate strength are in great agreement with the MD findings, demonstrating the predictive power of the ML approach.

We consider two additional MD simulations that employed strain rates lying beyond the training range, encompassing both extremes. These strain rates are defined as ${\dot{\epsilon }}_{3}=0.5\times 10^{-6}/fs$ (slower) and ${\dot{\epsilon }}_{4}=30\times 10^{-6}/fs$ (faster). The corresponding results are shown in Figure 10, together with the MD results, for comparison. Although the slower rate led to a slightly higher ultimate strength, remarkable agreement is observed within the elastic range. Here, the ANN effectively captured intricate aspects of the loading path. The findings clearly demonstrate the capacity of the model for generalization and robustness across various strain rates, which makes it a promising candidate for effectively describing the constitutive laws of crystalline polymers to be used in large-deformation FEM simulations.

In order to better quantify the model uncertainty of the ANN, we performed an error analysis. We use the same four sets of simulations with different strain rates as previously shown in Figure 9 and Figure 10. We consider the full six components of the prediction, namely the second Piola–Kirchhoff tensor $\hat{S}$, and investigate how the error evolves with the deformation by comparing it to the MD result S. Since the use of MSE does not make sense here, we use the percentage error defined below.
(12)$$\mathrm{Percentage}\;\mathrm{Error}=100\times \frac{||S-\hat{S}{\left|\right|}_{2}}{{\left||S|\right|}_{2}}\;,$$

Results are presented in Figure 11, with respect to the deformation. Note that we used the tensor component $C_{33}$ to represent axial deformation, to be consistent with the previous discussions. We observe that the error remains close to zero for the elastic region, except for the initial loading regime. When the material is further deformed and starts to yield, that is, for $C_{33}>1.20$, we start to see a significant increase in error. This is expected as the failure mechanism and the material response beyond yielding would involve high uncertainty. Out-of-domain data ${\dot{\epsilon }}_{3}$ result in a significant error, especially during the initial loading phase. On the other hand, ${\dot{\epsilon }}_{4}$ outcomes are similar to those of the in-range data. From these findings we infer that the model performs better when dealing with faster loading rates, and there is a high uncertainty within the initial loading and the failure regions. Further analysis may be required to draw a final conclusion as the data here were rather limited.

## 4. Conclusions

In this study, we have developed an artificial neural network to model the mechanical response of crystalline polymers based on molecular dynamics simulations. Initial configurations are generated to model PA12 in γ crystal form. ReaxFF was our choice for the force field, and MD simulations are performed under uniaxial and biaxial tensile deformations to generate a data set for the learning model.

ANN architecture was selected through hyperparameter tuning algorithms and carefully calibrated using the training set. The training and validation process showed the best model to be ANN-4, which was proven to be robust and accurate with no significant overfitting. The best model was tested against different strain rates, which covered above and beyond the loading rate range of the training set. The results demonstrated the predictive power and generalization capacity of our approach, even with strain rates that are much slower or faster than the original data.

The presented methodology proved to be an efficient way of modeling the microstructurally informed constitutive relation of crystalline polymers, which would be used in macroscale FEM simulations. Namely, we presented an approach to accurately predict the mapping between the right Cauchy–Green (C) and second PK2 stress (S) tensors, which shall be utilized to compute the strain energy density of large deformation problems. Once the model is trained, we can simply treat it as a black box for computing the strain energy at arbitrary quadrature points and time steps, corresponding to the current deformation state of the material.

There are some caveats that need further attention. As detailed in Section 2.3, there may be a mechanically induced phase transformation of PA12, which may depend on the particular additive manufacturing method and process parameters such as temperature and printing speed. Furthermore, we only considered uniaxial and biaxial tensile deformations in our training set. For general loading conditions, such as pure shear, compression, and torsion, the artificial neural network will need more data, and the associated ML training procedure may become more complicated and expensive. Nevertheless, the approach outlined in this work can easily be extended to any deformation mode by obtaining the corresponding MD data set and repeating the ML process.

It should be noted that our current investigation was limited to the fully crystalline form of PA12 for demonstration purposes. The actual material components consist of semicrystalline structures wherein the microstructure comprises amorphous and crystalline regions. However, the MD and ML methodologies presented can be easily adopted for amorphous domains. Additionally, considering larger simulation cells, having longer relaxation windows, and adopting slower strain rates may be necessary to simulate more accurate and realistic scenarios. These matters will be further investigated and reported in a separate study.

## Figures and Tables

**Figure 1 polymers-15-04254-f001:**
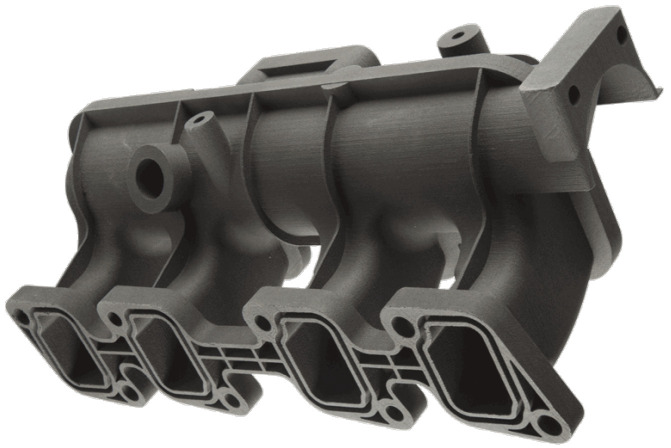
A 3D printed PA12 auto part via HP multi-jet fusion (Courtesy of Ford Motor Company).

**Figure 2 polymers-15-04254-f002:**
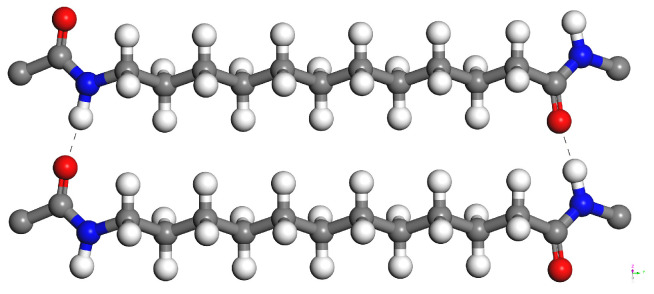
Polyamide12 molecules. Gray, white, blue, and red spheres represent carbon, hydrogen, nitrogen, and oxygen atoms, respectively. Dashed lines indicate the H bonds.

**Figure 3 polymers-15-04254-f003:**
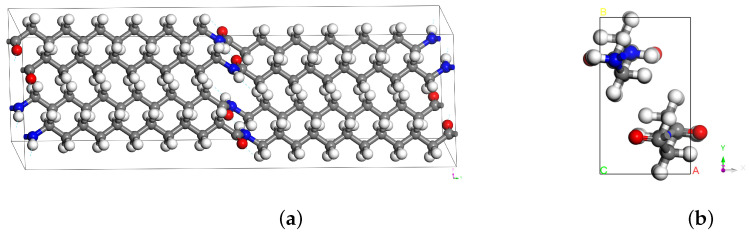
Unit cell of the γ form PA12: (**a**) monoclinic cell; (**b**) orthogonal transformation.

**Figure 4 polymers-15-04254-f004:**
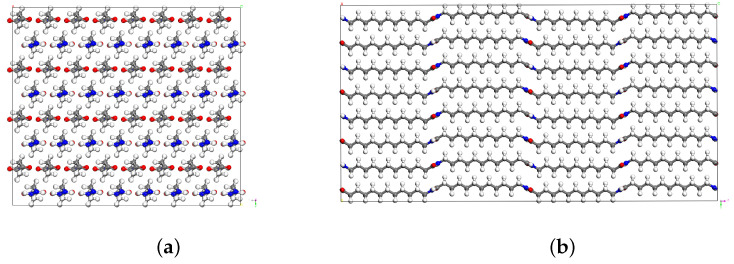
Supercell of perfect PA12 crystals: (**a**) view from x-y plane; (**b**) view from y-z plane.

**Figure 5 polymers-15-04254-f005:**
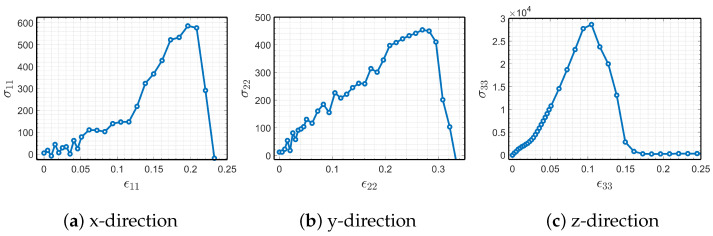
Stress (MPa) versus strain (engineering) behavior of PA12 in uniaxial tension.

**Figure 6 polymers-15-04254-f006:**
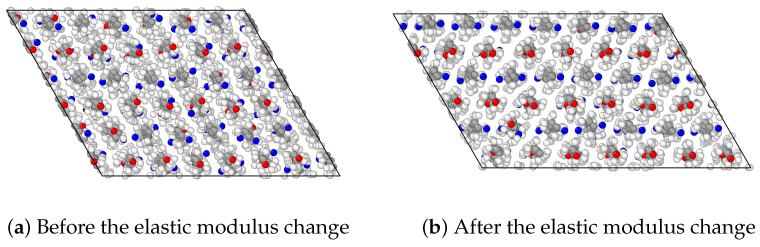
Snapshots of the atomistic configurations during deformation in the x-direction.

**Figure 7 polymers-15-04254-f007:**
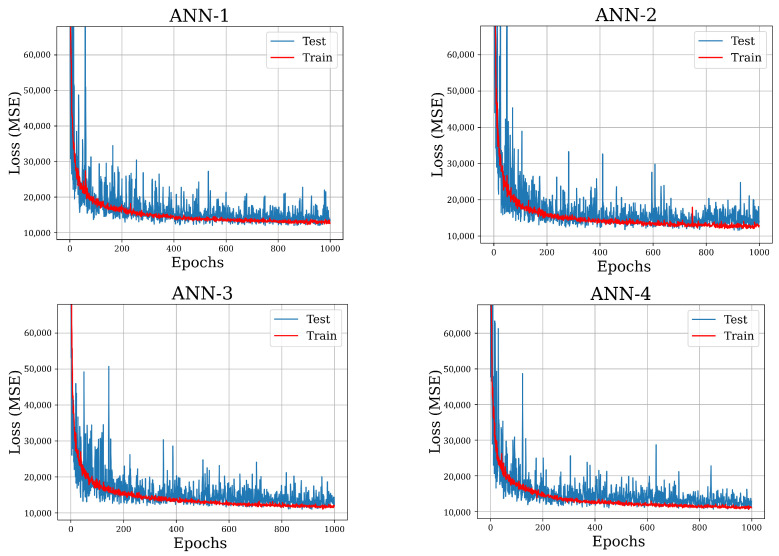
Learning curves of the ANNS: training and test loss (MPa) during training phase.

**Figure 8 polymers-15-04254-f008:**
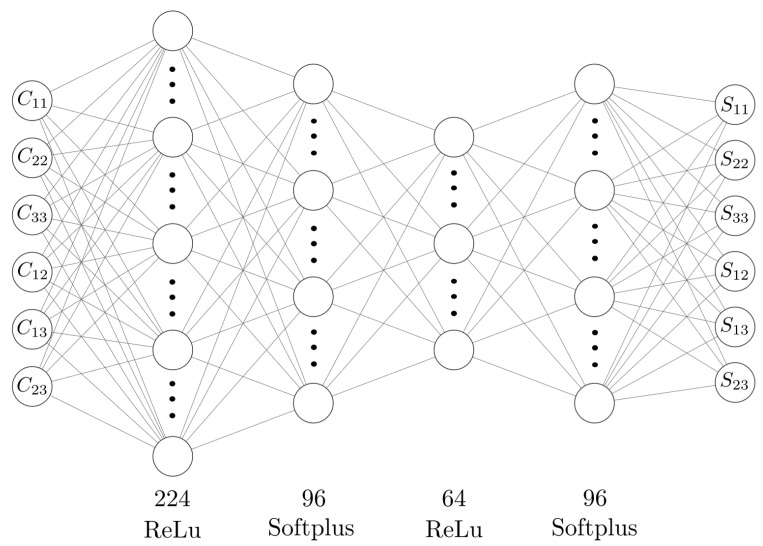
ANN-4 schematic representation [40].

**Figure 9 polymers-15-04254-f009:**
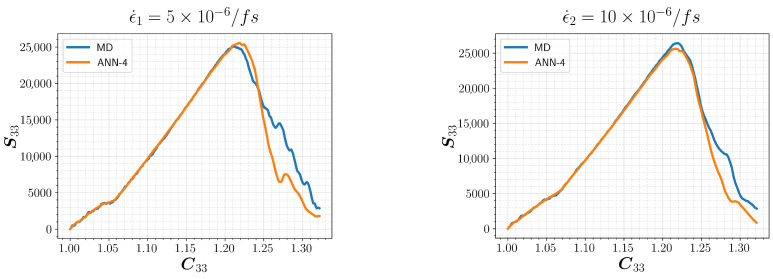
Predictions of ANN-4 for uniaxial tension in z-direction for a new set of simulations. The strain rates are contained in the training range.

**Figure 10 polymers-15-04254-f010:**
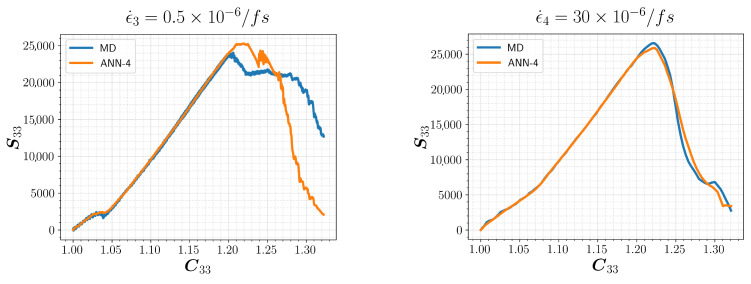
Predictions of ANN-4 for uniaxial tension in z-direction for a new set of simulations. The strain rates are beyond the range of the training data.

**Figure 11 polymers-15-04254-f011:**
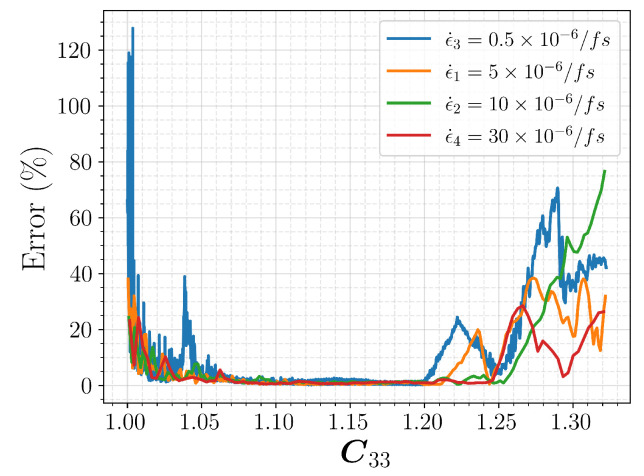
Percentage error, as defined in Equation (Equation 12), for different strain rates.

**Table 1 polymers-15-04254-t001:** Lattice parameters for γ form PA12 crystal from [13,14].

Lattice Parameters	
a	4.79 Å
b	31.90 Å
c	9.58 Å
β	120${}^{\circ }$
Space Group	P21/C

**Table 2 polymers-15-04254-t002:** Ultimate strain and resulting ranges of strain rates for each direction.

	Deformation Direction		
	**x**	**y**	**z**
Ultimate Strain	0.5	0.6	0.15
Strain Rate ($10^{-6}/fs$)	[3.3,50]	[4,60]	[1,15]

**Table 3 polymers-15-04254-t003:** Hyperparameter search grid.

Hyperparameters			
**Hidden Layers**	**Neurons**	**Activation Function**	**Learning Rate**
[2,5]	[32,256]	{ReLu, Softplus}	$\left\{10^{-1},10^{-2},10^{-3}\right\}$

**Table 4 polymers-15-04254-t004:** ANN architectures resulting from Hyperband method and Bayesian Optimization.

	Hyperband (1)	Bayesian (2)	Hyperband (3)	Bayesian (4)
Input Layer	6	6	6	6
Hidden Layer 1	192 × ReLu	64 × ReLu	160 × ReLu	224 × ReLu
Hidden Layer 2	32 × Softplus	32 × Softplus	64 × Softplus	96 × Softplus
Hidden Layer 3	32 × ReLu	256 × Softplus	32 × ReLu	64 × ReLu
Hidden Layer 4	64 × ReLu	32 × ReLu	128 Softplus	96 × Softplus
Hidden Layer 5	128 × ReLu	64 × ReLu	-	-
Output Layer	6 × Linear	6 ×Linear	6 × Linear	6 ×Linear
Validation Loss	15,630	15,026	14,109	13,538

## Data Availability

Data will be made available on request.

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
