# Peer review of "Artificial Neural Networks for Predicting Mechanical Properties of Crystalline Polyamide12 via Molecular Dynamics Simulations"

_polymers, 2023, doi:10.3390/polym15214254_

Round 1
Reviewer 1 Report
Comments and Suggestions for Authors
This study presents an artificial neural network (ANN) developed to predict the mechanical properties of Polyamide12 (PA12) in 3D printed products, using data from molecular dynamics simulations. The ANN accurately predicts PA12 stress-strain relations based on macroscale deformation, enabling multiscale modeling for semicrystalline polymers. However, before further consideration of the manuscript, the authors must “fully” address the comments listed below:
1- Why was it necessary to transform the simulation box into an orthogonal cell with chains aligned with the z-axis?
2- How were the damping parameters for the Nosé-Hoover thermostat and barostat selected for the MD simulations?
3- What is the significance of transforming the data from MD simulations into tensor quantities such as the right Cauchy-Green tensor and the second Piola-Kirchhoff stress tensor?
4- Could you explain the standardization process applied to the input data set for the ANN?
5- What are the practical applications and advantages of the ANN-4 model in predicting the mechanical response of crystalline polymers?
6- Discuss if you can expand the study to include other loading conditions beyond uniaxial and biaxial tensile deformations.
7- How does the presence of amorphous regions in semicrystalline polymers affect the mechanical properties? How sensitive are the ANN predictions to changes in the force field or model parameters?
8- Can you address the computational scalability of this approach for larger systems or longer simulations?
Reviewer 2 Report
Comments and Suggestions for Authors
Artificial neural networks for predicting mechanical properties of crystalline polyamide12 via molecular dynamics simulations
This manuscript investigates the feasibility of using a neural network model to predict the strain-stress relationship of crystalline Polyamide12 based on molecular dynamics simulations. This work firstly collects the training data by running deformation simulations with various loading conditions, e.g. ultimate strain, strain rate. Then systematically build NN models to infer the PK2 stress tensor based on Cauchy-Green tensor as input features. As a result, the learning curves of some typical models show this is a feasible methodology and the authors also show some out-of-domain cases regarding strain rates.
This work is mostly presented clearly and well written. In addition, ML-based mechanics is also a hot topic recently. I recommend accepting this manuscript after addressing the following questions.
-
Data collection: The description of MD simulations is somehow perplexing to me. Could you explain more clearly how you get constant ultimate strain along each direction with various strain rates given the same 10000 steps?
-
Feature engineering: Is stress response of Polyamide12 rate-dependent? If it is, why is the model able to recognize the strain-stress relationship given different loading rates while the input features only contain deformation gradients without having rate-related features?
-
Out-of-domain data: Comment 2 is also related to here - neural networks should have high model uncertainty when taking care of data from out-of-distribution data. Some typical cases as Figure 10 is not good enough to present the potential issue. Could you perform some error analysis to describe the actual performance? Such as how the MSE looks compared to one from holdout data.
Reviewer 3 Report
Comments and Suggestions for Authors
I have reviewed your manuscript, which presents a novel approach to predicting the mechanical properties of crystalline polyamide 12 (PA12) using ANN trained on data from molecular dynamics (MD) simulations. While the topic is relevant and the proposed method has potential, a major revision is necessary to address the following concerns:
1) The abstract lacks clarity and coherence. The main idea and the significance of the study are not well-defined. It's essential to provide a clear and concise introduction to the problem and the approach before discussing the results and implications.
2) In order to guarantee that your manuscript complies with the high standards of academic publishing, there are crucial matters that require careful consideration. It is in this context that I recommend a thorough revision, clarifying the following specified areas for improvement: Consider providing a stronger introduction using the following resources that will help readers understand the significance of the work.
"Dislocation motion in plastic deformation of nano polycrystalline metal materials: a phase field crystal method study"
"Evolution of crystallographic orientation, precipitation, phase transformation and mechanical properties realized by enhancing deposition current for dual-wire arc additive manufactured Ni-rich NiTi alloy"
"Fresh, mechanical and microstructural properties of alkali-activated composites incorporating nanomaterials: A comprehensive review"
"Unraveling of Advances in 3D-Printed Polymer-Based Bone Scaffolds"
"Friction behavior of biodegradable electrospun polyester nanofibrous membranes"
3) The discussion section should interpret the results in a broader context. How does the proposed approach compare to existing methods for predicting material properties in additive manufacturing?
4) How does the ANN handle missing or incomplete data, if applicable?
5) How can the model be extended or improved for a broader range of semicrystalline polymers or different materials?
6) The choice of using the right Cauchy-Green tensor and the second Piola-Kirchhoff stress tensor for representing the deformation state and material response is mentioned. Can you discuss the reasons for selecting these particular tensors and how they are utilized in your modeling approach?
7) Considering the limitations mentioned, such as high strain rates and short simulation times, how did you ensure the reliability and representativeness of the data set for modeling the constitutive law? Were any sensitivity analyses performed to assess the impact of these limitations on the model's accuracy?
Comments on the Quality of English LanguageThe English structure of the paper should be checked carefully.
Round 2
Reviewer 1 Report
Comments and Suggestions for Authors
comments are addressed and the manuscript can be published.
Reviewer 3 Report
Comments and Suggestions for Authors
After reviewing the revised version of your manuscript, I am happy to inform you that the paper can now be accepted for publication.